

# GHOST: Geoscientific Hollow Sphere Tesselation

Cedric Thieulot[1]

[1]Utrecht University, The Netherlands

*Correspondence to:* C. Thieulot (c.thieulot@uu.nl)

**Abstract.** I present in this work the GHOST (Geoscientific HOllow Sphere Tesselation) software which allows for the fast generation of computational meshes in hollow sphere geometries counting up to a hundred millions of cells. Each mesh is composed of concentric spherical shells which are built out of quadrilaterals or triangles. I focus here on three commonly used meshes used in geodynamics/geophysics and demonstrate the accuracy of shell surfaces and mesh volume measurements as a function of resolution. I further benchmark the built-in gravity and gravitational potential procedures in the simple case of a constant density geometry and finally show how the produced meshes can be used to visualise the S40RTS mantle tomography model. The code is open source and is available on the GitHub sharing platform.

## 1 Introduction

In the last 40 years numerical mantle convection studies have improved our understanding of mantle dynamics as a whole (Schubert et al., 2001). While early studies looked at aspects of fluid dynamics aspects (Busse, 1975; Christensen and Harder, 1991), more recent studies have been exploring a wide variety of topics. For example mantle mixing (van Keken et al., 2002), melting (Tackley, 2012; van Heck et al., 2016; Dannberg and Heister, 2016), the effect of plate motion history on the longevity of deep mantle heterogeneities (Bull et al., 2014), or assimilating lithosphere and slab history in 4-D Earth models (Bower et al., 2015).

To a first approximation the Earth is a sphere: the Earth's polar diameter is about 43 kilometers shorter than its equatorial diameter, a negligeable difference of about 0.3%. As a consequence, modelling physical processes which take place in the planet require the discretisation of a sphere. Furthermore, because core dynamics occur on vastly difference time scales than mantle dynamics, mantle modelling usually leaves the core out, thereby requiring simulations to be run on a hollow sphere mesh.

Although so-called latitude-longitude grids would seem appealing, they suffer from the convergence of meridians at the poles (resulting in over sampling at poles) and the juxtaposition of triangles near the poles and quadrilaterals elsewhere. As a consequence more regular, but more complex, grids have been designed over the years which tesselate the surface of the sphere into triangles or quadrilaterals (sometimes overlapping) There is the 'cubed sphere' (Ronchi et al., 1996; Choblet et al.,



2007), the ying-yang grid (Kageyama and Sato, 2004; Yoshida and Kageyama, 2004; Kameyama et al., 2008; Tackley, 2008; Crameri and Tackley, 2014, 2016), the spiral grid (Hüttig and Stemmer, 2008), an icosahedron-based grid (Baumgardner and Frederickson, 1985; Tabata and Suzuki, 2002), or a grid composed of 12 blocks further subdivided into quadrilaterals (Zhong et al., 2000) as used in the CitcomS code.

How such meshes are built is often not discussed in the literature and while it is a simple exercise of three-dimensional geometry it can be time-consuming, especially the connectivity array generation. In this paper I present a simple open source mesh generator for three hollow sphere meshes: the 'cubed sphere' mesh, the CitcomS mesh and the icosahedral mesh.

    I first present the basic workflow which has been implemented to arrive at such meshes, then I showcase its efficiency and how accurate surfaces and volumes are represented. Finally, I provide a simple example of gravity and gravity potential

calculations on such meshes and compare the obtained values with the analytical solution derived in the Appendix.

## 2   Building the hollow sphere meshes

The open source code library GHOST allows three different types of hollow sphere meshes to be built , i.e. meshes bounded by two concentric spheres:

– The cubed sphere ('HS06'), composed of 6 blocks which are themselves subdivided into $N_b \times N_b$ quadrilateral shaped

cells (Sadourny, 1972; Ronchi et al., 1996; het, 2003; Burstedde et al., 2013). Four types of cubed spheres meshes have been proposed: the conformal, elliptic, gnomonic and spring types (Putman and Lin, 2007). However only gnomonic meshes are considered here: these are obtained by inscribing a cube within a sphere and expanding to the surface of the sphere. The cubed sphere has recently been used in large-scale mantle convection simulation in conjunction with Adaptive Mesh Refinement (Alisic et al., 2012; Burstedde et al., 2013).

– The CitcomS mesh ('HS12') composed of 12 blocks also subdivided into $N_b \times N_b$ quadrilateral shaped cells (Zhong et al., 2000; Stemmer et al., 2006; Zhong et al., 2008; Arrial et al., 2014). Note that ASPECT (Kronbichler et al., 2012; Heister et al., 2017), a relatively new code aimed at superseeding CitcomS can generate and use this type of mesh (Thieulot, 2017) but is not limited to it.

– The icosahedral mesh ('HS20') composed of 20 triangular blocks (Baumgardner and Frederickson, 1985; Baumgardner,

1985) subdivided into triangles, which is used in the TERRA code (Bunge et al., 1996, 1997, 1998; Davies et al., 2013).

    Given the regularity and symmetry of these meshes determining the location of the mesh nodes in space is a relatively straightforward task. Building the mesh connectivity in an efficient manner is where the difficulty lies.

    The approach to building all three meshes is identical:

1. A reference square or triangle is populated with cells, as shown in Fig. (1) parametrised by a level $l$: the square is

subdivided into $l \times l$ quadrilaterals while the triangle is subdivided into $l^2$ triangles.



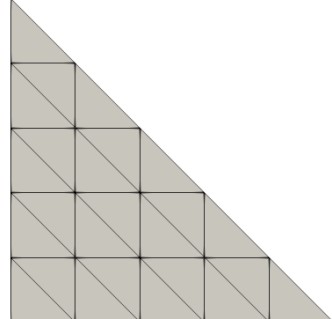

**Figure 1.** Reference square and triangles meshes at level 5

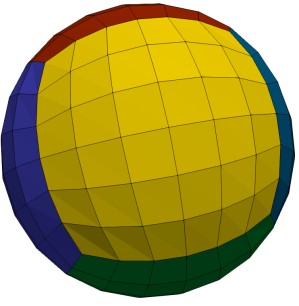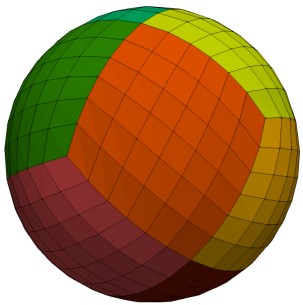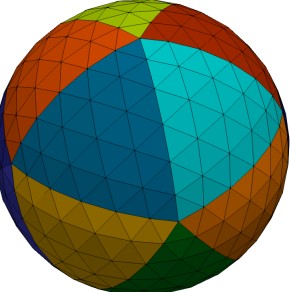

**Figure 2.** From left to right: HS06, HS12 and HS20 shells coloured by block number.

2. This reference square or triangle is then replicated *nblock* times (6, 12 or 20) and mapped onto a portion of a unit sphere. The blocks are such that their union covers a full sphere but they cannot overlap except at the edges, see Fig. (2).

3. All block meshes are then merged together to generate a shell mesh.

4. Shell meshes are replicated *nlayer+1* times outwards with increasing radii.

5. The *nlayer* shells are then merged together to form a hollow sphere mesh, as shown in Fig. (3).

In Table (1) the number of nodes and cells for a variety of resolutions for all three mesh types is reported. Looking at the CitcomS literature of the past 20 years, we find that the mesh data presented in this table cover the various resolutions used, e.g. $12 \times 48^3$ (McNamara and Zhong, 2004; Arrial et al., 2014), $12 \times 64^3$ (Bull et al., 2014) $12 \times 96^3$ (Bull et al., 2010), $12 \times 128^3$ (Becker, 2006; Weller and Lenardic, 2016; Weller et al., 2016). Note that in the case of the HS06 and HS12 meshes the mesh nodes are mapped out to the 6 or 12 blocks following either an equidistant or equiangle approach as shown in Fig. (6) (see Putman and Lin (2007) for details on both approaches).



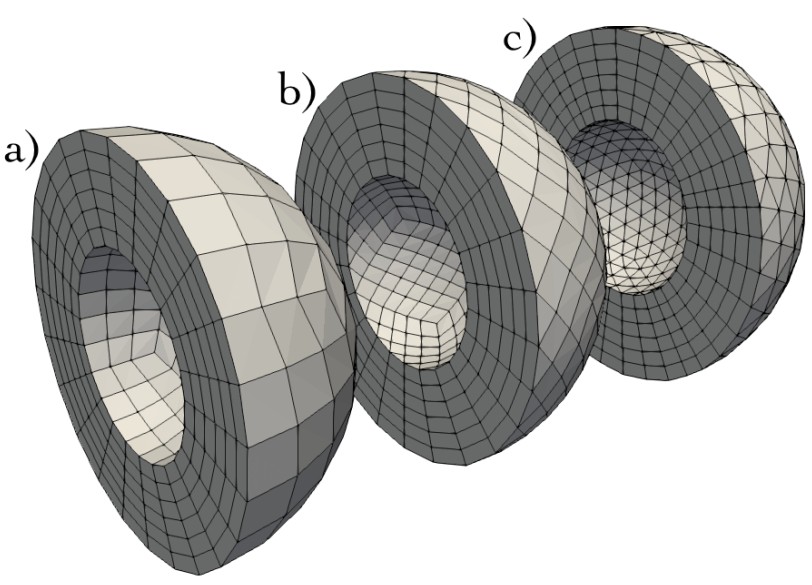

**Figure 3.** a) HS06 mesh composed of 6 blocks containing each $6^3$ cells; b) HS12 mesh composed of 12 blocks containing each $6^3$ cells; e) HS20 mesh composed of 20 blocks containing each $6^3$ cells.

## 2.1 Mesh generation performance

The total time to generate the coordinates and the connectivity of the final hollow sphere mesh was timed for all three mesh types and is shown in Fig. (4). The reported times scale linearly with the number of nodes up to 100 million mesh nodes and a mesh containing a million or so nodes can be built in less than a second on a laptop.

5 ## 2.2 Areas and volume measurements

The area of the shell of unit radius can be calculated by summing the areas of each cell. In the case of triangles Heron's formula is used, which states that the area of a triangle whose sides have lengths $a$, $b$, and $c$ is

$$A = \sqrt{s(s-a)(s-b)(s-c)} \qquad (1)$$

where s is the semiperimeter of the triangle:

10  $$s = (a+b+c)/2 \qquad (2)$$





| type | level | N | Nel | structure |
|------|-------|---|-----|-----------|
| HS06 | 2 | 78 | 48 | $6 \times 2^3$ |
| HS06 | 4 | 490 | 384 | $6 \times 4^3$ |
| HS06 | 8 | 3,474 | 3,072 | $6 \times 8^3$ |
| HS06 | 16 | 26,146 | 24,576 | $6 \times 16^3$ |
| HS06 | 32 | 202,818 | 196,608 | $6 \times 32^3$ |
| HS06 | 64 | 1,597,570 | 1,572,864 | $6 \times 64^3$ |
| HS06 | 128 | 12,681,474 | 12,582,912 | $6 \times 128^3$ |
| HS06 | 256 | 101,057,026 | 100,663,296 | $6 \times 256^3$ |
| HS12 | 2 | 150 | 96 | $12 \times 2^3$ |
| HS12 | 4 | 970 | 768 | $12 \times 4^3$ |
| HS12 | 8 | 6,930 | 6,144 | $12 \times 8^3$ |
| HS12 | 16 | 52,258 | 49,152 | $12 \times 16^3$ |
| HS12 | 32 | 405,570 | 393,216 | $12 \times 32^3$ |
| HS12 | 48 | 1,354,850 | 1,327,104 | $12 \times 48^3$ |
| HS12 | 64 | 3,195,010 | 3,145,728 | $12 \times 64^3$ |
| HS12 | 128 | 25,362,690 | 25,165,824 | $12 \times 128^3$ |
| HS12 | 256 | 202,113,538 | 201,326,592 | $12 \times 256^3$ |
| HS20 | 2 | 126 | 160 | $20 \times 2^3$ |
| HS20 | 4 | 810 | 1,280 | $20 \times 4^3$ |
| HS20 | 8 | 5,778 | 10,240 | $20 \times 8^3$ |
| HS20 | 16 | 43,554 | 81,920 | $20 \times 16^3$ |
| HS20 | 32 | 337,986 | 655,360 | $20 \times 32^3$ |
| HS20 | 64 | 2,662,530 | 5,242,880 | $20 \times 64^3$ |
| HS20 | 128 | 21,135,618 | 41,943,040 | $20 \times 128^3$ |
| HS20 | 256 | 168,428,034 | 335,544,320 | $20 \times 256^3$ |

**Table 1.** Number of nodes and elements/cells for the three types of meshes and for various levels.

In the case of quadrilaterals the four points composing each of them are not necessarily coplanar and the definition of the surface is ill-posed. Each quadrilateral is therefore decomposed into four triangles sharing a vertex in the middle given by the barycenter of the four points. The area of each quadrilateral is then approximated by the sum of the areas of all four inscribed triangles.

5    The relative error on the shell area is shown in Fig. (5) and for all three meshes the error is found to decrease linearly with the number of points for all three types of meshes.

Fig. (6) shows shell which approximately count the same number of nodes. We see that the equiangle projection of the nodes for the HS06 and HS12 meshes yields cells whose area are homogenous in value than when the equidistant projection is used.





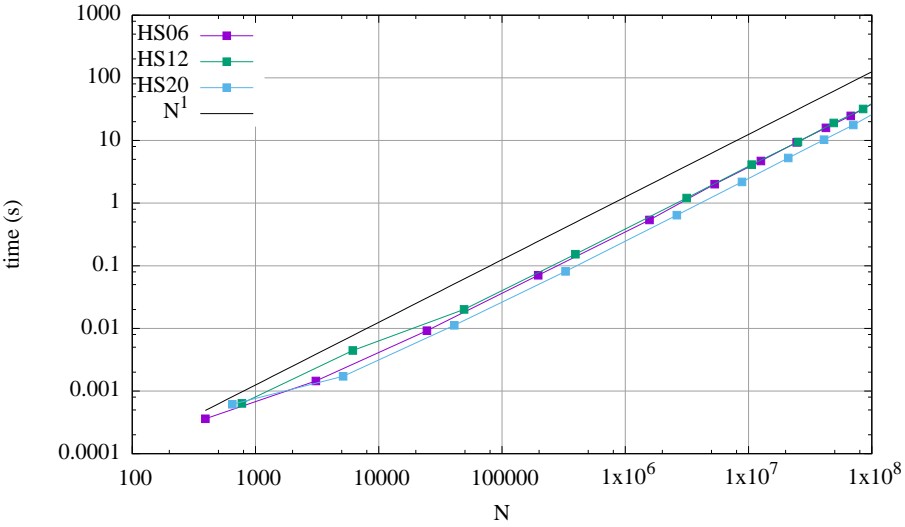

**Figure 4.** Measured times to build and assemble the mesh as a function of its number of nodes

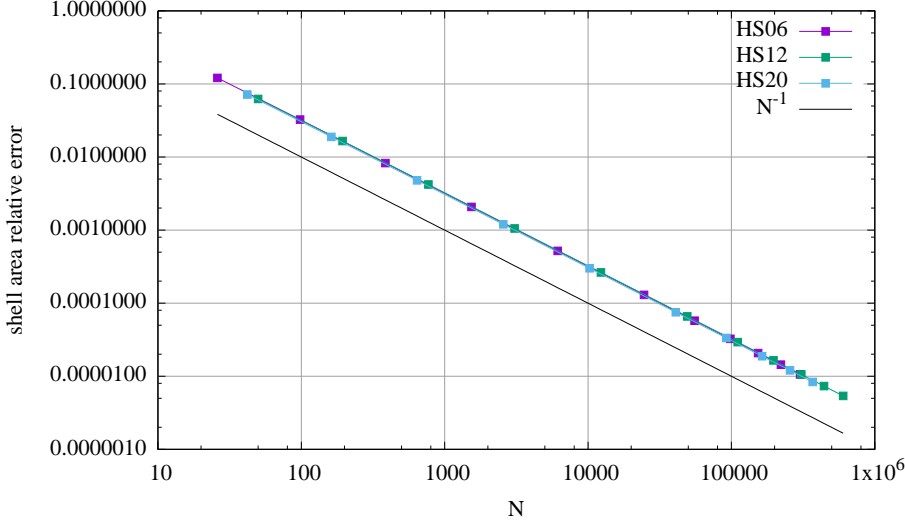

**Figure 5.** Relative shell area error as a function of its number of nodes

Although the volume of the hexahedra could have been computed directly with the formula of Grandy (1997), with GHOST the Gauss quadrature is used since it is also needed in the next section. The total volume of the spherical mesh is given by:

$$V = \int\!\!\int\!\!\int_{\Omega} dxdydz = \sum_c \int\!\!\int\!\!\int_{\Omega_c} dxdydz \tag{3}$$


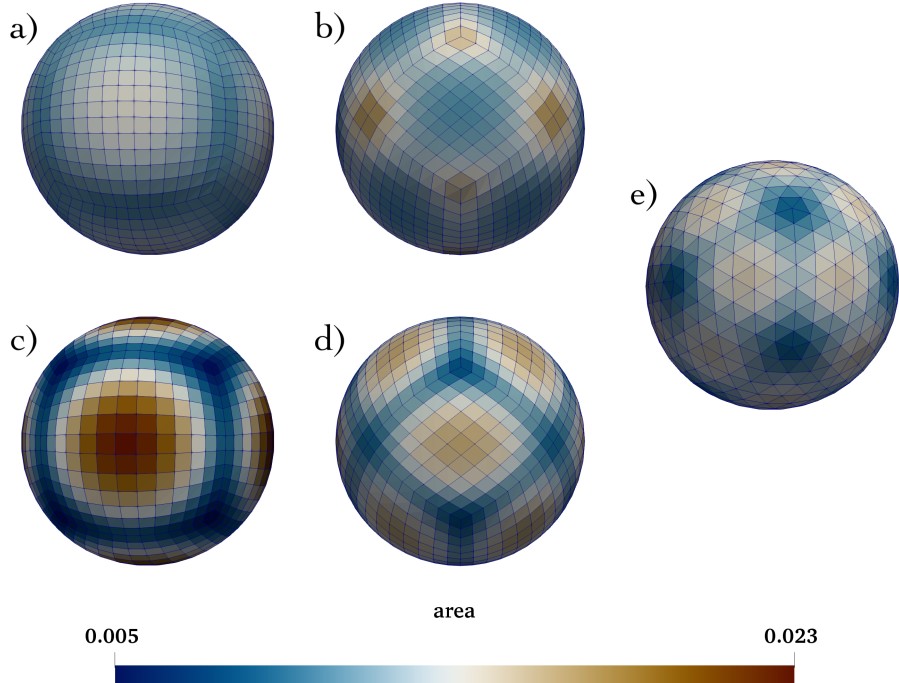

**Figure 6.** Elemental area for a) equiangular HS06 mesh, b) equiangular HS12 mesh, c) equidistant HS06 mesh d) equidistant HS12mesh e) HS20 (level 7)

where $\Omega$ stands for the volume inside the spherical shell, $\Omega_c$ is the volume of a cell. The sum runs over all the cells and a $2 \times 2 \times 2$ quadrature rule is used in each cell.

The relative error on the hollow sphere mesh volume is shown in Fig. (7) and for all three meshes the error is found to decrease with the number of points for all three types of meshes with a $-2/3$ exponent.

5   ## 3   Gravity field and potential measurements

The gravity potential can be computed by means of the Poisson equation $\boldsymbol{\nabla}^2 U = 4\pi \mathcal{G} \rho$ where $\mathcal{G}$ is the gravitational constant (see Turcotte and Schubert, 2012, Chapt. 5). The analytical solution of this equation in the case of a constant density spherical shell is given in Appendix A.

In what follows I set $R_1 = 1$, $R_2 = 2$, $\rho_0 = 10^6$ and $\mathcal{G} = 6.673848 \times 10^{-11} \mathrm{m}^3 \, \mathrm{kg}^{-1} \, \mathrm{s}^{-2}$. Figs. (8) and (9) show the gravity
10   $g = |\boldsymbol{g}|$ and potential $U$ measured at 256 points along a line between $r = 0$ and $r = 4R_2$ for all three mesh types. Data points align along the analytical curves and the error is found to decrease as a function of the mesh resolution.





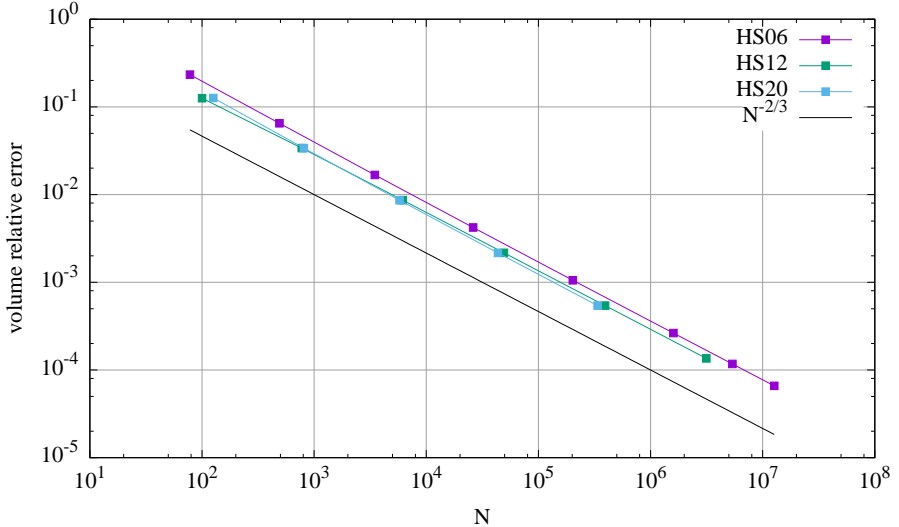

**Figure 7.** Volume relative error as a function of its number of nodes for all three mesh types.

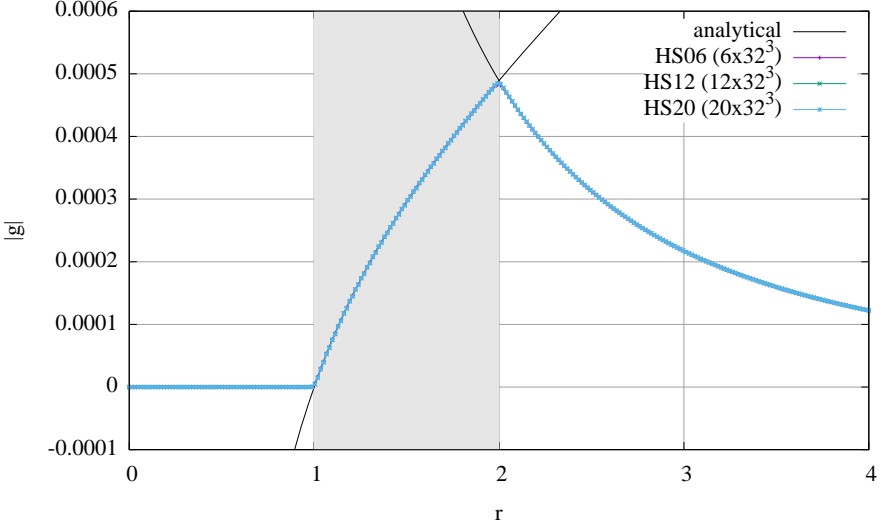

**Figure 8.** Norm of the gravity vector $|\boldsymbol{g}|$ as a function of the radial coordinate $r$. The gray area symbolises the spherical shell.

## 4   Application: visualisation of a tomography dataset

The S40RTS model is one of the most widely used tomographic models of the mantle (Ritsema et al., 2011). It is based on 20 million Rayleigh wave dispersion, 500,000 shear-wave Traveltime, and 1100 normal-mode Splitting function measurements.





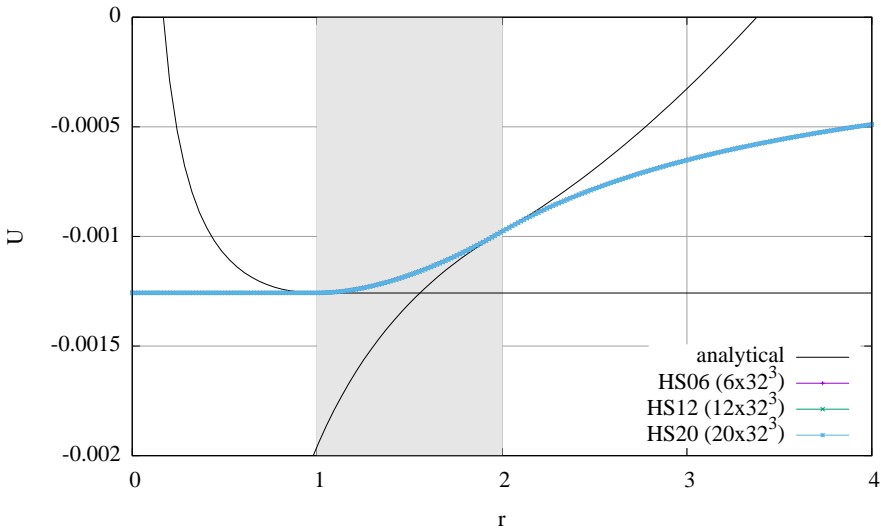

**Figure 9.** Gravity potential $U$ as a function of the radial coordinate $r$. The gray area symbolises the spherical shell.

The data is widely available, for instance on the website of the main author (http://jritsema.earth.lsa.umich.edu//Research.html) or as part of the SPECFEM 3Dglobe code (https://github.com/geodynamics/specfem3d_globe).

I have written a simple interface to the dataset and for each node of the grid the shear-velocity variation $\delta \ln V_s$ is computed, as well as the relative density variation $\delta \ln \rho = \xi \delta \ln V_s$ where $\xi = 0.25$ is assumed to be constant with depth for simplicity (see

Fig. 6 of Steinberger and Calderwood (2006)). The absolute density variation with regards to the PREM model (Dziewonski and Anderson, 1981) is then obtained as follows: $\delta \rho = \rho_{PREM} * \delta \ln \rho$. Results are shown in Fig. (10)

## 5   Conclusions

The three types of hollow sphere meshes presented in this work are currently in use in the ELEFANT code (http://cedricthieulot.net/elefant.h Furthermore the HS12 mesh was recently used in Thieulot (2017) in which a family of analytical solutions for viscous incom-

pressible Stokes flow in a spherical shell is presented.

Following the example of CitcomS, each block of the final mesh could actually be built and used by a different MPI thread in the context of parallel calculations Burstedde et al. (2013). Each block could then subsequently be divided to allow for more threads to be used than the original number of blocks.

Other tomography models than S40RTS (Ritsema et al., 2011) could have been chosen such as UUP07 (van der Meer

et al., 2017; Hall and Spakman, 2015) and other geophyical databases could have been coupled with it, such as the crust and lithospheric model Litho1.0 (Pasyanos et al., 2014) to arrive at a more complete high resolution of the Earth. Gravity (anomaly) and geoid measurements could then be carried out.

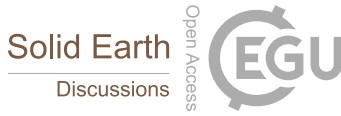



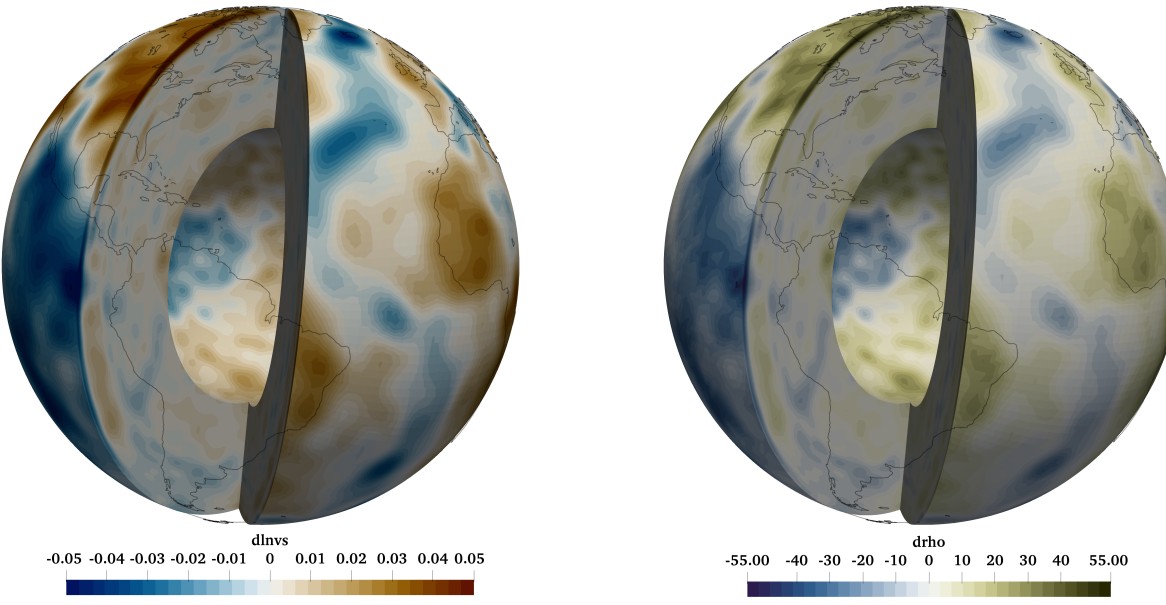

**Figure 10.** HS06 grid with 64 layers and level=64. a) $\delta \ln V_s$ at the surface of the model ($R = 6346$km) and at depth down to the core mantle boundary ($R = 3480$km); b) absolute density variation $\delta \rho$ (kg/m$^3$).

Finally, this library is aimed at students and researchers alike. It provides the essential building block for many geophysical applications: a non overlapping tesselation of the mantle. It also provides the starting point for any Finite Element or Finite Volume Method based geodynamical code.

*Code availability.* The code is written in Fortran90 and is freely downloadable at `https://github.com/cedrict/GHOST`.

# 5 Appendix A: Analytical solution for the gravity and gravitational potential fields inside and outside a constant density spherical shell

The gravity potential can be computed by means of the Poisson equation $\boldsymbol{\nabla}^2 U = 4\pi\mathcal{G}\rho$ where $\mathcal{G}$ is the gravitational constant (Turcotte and Schubert, 2012). The density is non zero only for $R_1 \leq r \leq R_2$. Outside the spherical shell one then needs to solve the Laplace equation $\Delta U = 0$ which simplifies to:

$$10 \quad \frac{1}{r^2}\frac{\partial}{\partial r}\left(r^2\frac{\partial U}{\partial r}\right) = 0 \tag{A1}$$

by symmetry which has the simple solution:

$$g = \frac{\partial U}{\partial r} = \frac{C}{r^2} \tag{A2}$$





where $C$ is a constant. In order to avoid an infinite gravity field at $r = 0$, we need to impose $C = 0$, i.e. the gravity is zero for $r <= R_1$. Inside the shell, $\rho = \rho_0$ and we easily obtain:

$$g = \frac{\partial U}{\partial r} = \frac{4\pi}{3}\mathcal{G}\rho_0 r + \frac{A}{r^2} \tag{A3}$$

where $A$ is an integration constant. We know that $g = 0$ at the inner boundary $r = R_1$ (no mass within a radius $r \leq R_1$ so we can compute $A$ and finally:

$$g = \frac{\partial U}{\partial r} = \frac{4\pi}{3}\mathcal{G}\rho_0(r - \frac{R_1^3}{r^2}). \tag{A4}$$

The branch for $r \geq R_2$ is given by Eq. (A2) and requiring the gravity field to be continuous at $r = R_2$:

$$g(r) = \frac{\mathcal{G}M}{r^2} \tag{A5}$$

where $M = \frac{4\pi}{3}\pi\rho_0(R_2^3 - R_1^3)$ is the mass contained in the shell. Turning to the potential, we obtain its expression for $r >= R_2$ by integrating Eq.(A5):

$$U(r) = -\frac{\mathcal{G}M}{r} + D \tag{A6}$$

where $D$ is an integration constant which has to be zero since we require the potential to vanish for $r \to \infty$.

For $R_1 \leq r \leq R_2$, Eq. (A4) yields:

$$U(r) = \frac{4\pi}{3}\mathcal{G}\rho_0(\frac{r^2}{2} + \frac{R_1^3}{r}) + F \tag{A7}$$

where $F$ is a constant. Continuity of the potential at $r = R_2$ requires that

$$F = -2\pi\rho_0\mathcal{G}R_2^2. \tag{A8}$$

Since gravity is zero for $r \leq R_1$ the potential is then constant and continuity requirements yield

$$U(r) = 2\pi\mathcal{G}\rho_0(R_1^2 - R_2^2). \tag{A9}$$

*Competing interests.* The author declares that he has no conflict of interest.

*Acknowledgements.* The author wishes to thank A. Plunder and S. Boulay for stimulating discussions in the early stages of this work and H. Brett for careful reading of the manuscript. Perceptually-uniform colour maps were used in this study to prevent visual distortion of the data (http://www.fabiocrameri.ch/). Data visualisation is carried out with the Paraview software (http://paraview.org/).



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
