# Peer review of "GHOST: Geoscientific Hollow Sphere Tesselation"

_Solid Earth, 2018_

## Referee Comment (RC1) · F. Crameri (Referee) · 28 Mar 2018

GENERAL COMMENTS

The manuscript summarises the methodology behind building computational meshes for spherical shells and provides a useful, novel open-source software to build three commonly used 'hollow sphere' meshes efficiently. Even though there are many numerical codes built upon such meshes, I am not aware of any comparable open-source tool to built these meshes from scratch. As such it will proof useful across many numerical modelling disciplines in Earth Sciences and for students and advanced researchers alike. The manuscript is concise, well structured and clearly presented. I only have a few suggestions that might make the current manuscript even more useful to interested readers, and a couple of minor specific comments.

SPECIFIC COMMENTS

[Figure]

The manuscript explains the theory behind the code GHOST, but does not explain how to use it in practise. I think it would be useful to most interested readers if the author would consider adding a short section about the actual use of the code. Maybe something similar to section 3 of the manual that is provided with the code itself (which should also be referred to in the manuscript).

The manuscript provides a nice comparison between three different spherical shell grids, but a conclusive discussion about which grid performs best (which might depend on specific circumstances) is not provided (I guess a grid spacing that is as equal as possible is one criterium). Interested readers might find it useful to read about the author's conclusion on that maybe in the discussion section.

The author could also consider ensuring the long-term availability of the code by providing a DOI to the code itself (or to a specific version). Zenodo (https://zenodo.org) has, for example, an option to easily link an existing GitHub account and provide a DOI. Also, citing a number (in the title or text) for the specific code version discussed in the manuscript might be helpful.

TECHNICAL CORRECTIONS

page 1: lines 19-20: Consider rewording this slightly to account for the possibility to model a spherical planet in a rectangular 3-D grid. Even though this might cause some numerical artefacts, it is possible after all, and has been used in the past. line 24: Full stop missing after the bracket.

page 5: line 7: Is there something missing grammatically in this sentence? line 8: same here: the sentence does not seem to make sense grammatically.

page 7: lines 2-3: define variables c,dx,dy,dz. lines 3-4: consider clarifying that the error decreases "with increasing number of points". line 6: define rho line 9: what is rho_0?

appendix A: general: declare all variables used; e.g., rho, r, R1, R2, . . .

table 1: consider declaring 'N' and 'Nel' in the caption, and clarifying the three different grid type acronyms.

figures 4, 5, 7: for clarification, define what variable 'N' is and that the tags 'HSxx' are the three grid types.

figure 5 & 7: consider to clarify whether the error is percentage of total volume or something else.

figure 6: commas and full stop are missing in the caption.

figure 7: declare 'N' in the caption.

Fabio Crameri 28.03.2018

---

## Referee Comment (RC2) · C. Hüttig (Referee) · 3 May 2018

GENERAL

The paper presents open-source software that generates radially projected spherical shell grids, used typically in geophysical applications. Although no novel scientific method is presented, the code will serve well as an exercise for everyone starting with numerical simulations in this field.

Unfortunately the source code is only in Fortran, many other modern languages would benefit from such a library as well.

SPECIFIC COMMENTS

As with every algorithm, complexity is key. From your timing plot in Fig 4 it looks like linear complexity $O(n)$, which would be ideal. If so please make a statement.

[Figure]

In your introduction you highlight the difficulty to derive the connectivity between cells. I agree and would like to see a chapter on how you tackle this. Another quite similar topic is avoiding duplicate vertices in recursive building algorithms.

For numerical simulations it is often important that cells stay as geometrically constant as possible to avoid introducing errors based on mesh irregularities. As you have figured out, a good measure for this error is the area variance of the cells within a shell. Unfortunately you plotted this error in a way that makes selecting the "best" method concerning this error impossible. Also, why the volumetric relative error is of importance in a projected scenario (Fig.7) puzzles me.

I do not understand what you do in chapter 3. Comparing analytical solutions to numerical results is always a good idea. But how do you solve U? What method, order, ... The nature of your problem suggests a spectral method as others would struggle with the asymptotic boundary condition, please elaborate. Also, as the absolute error within a single shell (lateral only, for i.e. the middle shell) for each cell is interesting in this scenario as it also reflects the sensitivity of the numerical method to mesh irregularities.

In chapter 4 the application is quite useful, but you seem to describe an interpolation method with your statement "I have written a simple interface . . ." . Please describe in detail how you interpolate the data onto the mesh (generic algorithm based on connectivity or specific algorithms for each grid type? order of interpolation?). Do not use the word simple.

All the best,

Christian Hüttig

---

## Author Comment (AC1) · 21 May 2018

Dear Reviewer,

thank you for your positive feedback.

I hereunder reply to your 3 main points:

*The manuscript explains the theory behind the code GHOST, but does not explain how to use it in practice. I think it would be useful to most interested readers if the author would consider adding a short section about the actual use of the code. Maybe something similar to section 3 of the manual that is provided with the code itself (which should also be referred to in the manuscript).*

I agree and will add a section in the manual giving an example of how one could use it.

[Figure]

*The manuscript provides a nice comparison between three different spherical shell grids, but a conclusive discussion about which grid performs best (which might depend on specific circumstances) is not provided (I guess a grid spacing that is as equal as possible is one criterion). Interested readers might find it useful to read about the author's conclusion on that maybe in the discussion section.*

I agree there too and will add a small paragraph on this topic in the discussion section.

*The author could also consider ensuring the long-term availability of the code by providing a DOI to the code itself (or to a specific version). Zenodo (https://zenodo.org) has, for example, an option to easily link an existing GitHub account and provide a DOI. Also, citing a number (in the title or text) for the specific code version discussed in the manuscript might be helpful.*

I have generated a v1.0 release and I have linked my github account to Zenodo. This version has been attributed a DOI and will be explicitly mentioned in the manual and the revised article. https://doi.org/10.5281/zenodo.1245533

Best regards,

Cedric.

---

## Author Comment (AC2) · 21 May 2018

Dear Reviewer,

thank you for your positive and constructive feedback.

I hereunder address the points you raised:

*The paper presents open-source software that generates radially projected spherical shell grids, used typically in geophysical applications. Although no novel scientific method is presented, the code will serve well as an exercise for everyone starting with numerical simulations in this field. Unfortunately the source code is only in Fortran, many other modern languages would benefit from such a library as well.*

This is of course a valid remark but had I written it in Python or C++ such a remark would remain since every modern language has a different way of dealing with objects

and would require a different process to interface GHOST to whatever application is being written by the user. My philosophy is that fortran is a very readable language and either users will be content with the existing interface for their own application, either they will adapt (if not entirely translate) the code I provide to suit their own needs.

*As with every algorithm, complexity is key. From your timing plot in Fig 4 it looks like linear complexity O(n), which would be ideal. If so please make a statement.*

This is a good remark indeed and I will add a statement in the text.

*In your introduction you highlight the difficulty to derive the connectivity between cells. I agree and would like to see a chapter on how you tackle this. Another quite similar topic is avoiding duplicate vertices in recursive building algorithms.*

It was indeed one of the hardest algorithms to implement. I will add a paragraph to the revised version about this topic.

*For numerical simulations it is often important that cells stay as geometrically constant as possible to avoid introducing errors based on mesh irregularities. As you have figured out, a good measure for this error is the area variance of the cells within a shell. Unfortunately you plotted this error in a way that makes selecting the "best" method concerning this error impossible. Also, why the volumetric relative error is of importance in a projected scenario (Fig.7) puzzles me.*

This is a good point. I will better document the area variance of the cells for each grid in the revised version.

*I do not understand what you do in chapter 3. Comparing analytical solutions to numerical results is always a good idea. But how do you solve U? What method, order, . . . The nature of your problem suggests a spectral method as others would struggle with the asymptotic boundary condition, please elaborate. Also, as the absolute error within a single shell (lateral only, for i.e. the middle shell) for each cell is interesting in this scenario as it also reflects the sensitivity of the numerical method to mesh*

*irregularities.*

I use an integral equation, and not the Laplace form of the text. As such no PDE is solved, only a domain integral I use a simple 'brute force' approach by looping over all elements/cells and using 2x2x2 Gauss quadrature points to compute each cell integral (see manual for compute_gravity_ at_point subroutine - section 4.7). I will add a line in the revised version to clarify this.

*In chapter 4 the application is quite useful, but you seem to describe an interpolation method with your statement "I have written a simple interface . . .". Please describe in detail how you interpolate the data onto the mesh (generic algorithm based on connec- tivity or specific algorithms for each grid type? order of interpolation?). Do not use the word simple.*

I agree that the statement 'simple interface' is vague and I will clarify this in the revised version. I am actually making use of the provided subroutine coming with the S40RTS dataset which return the $\delta \ln v_s$ at any point.

Best regards,

Cedric.

---

## Author Response (AR2)

**1 Reviewer1**

*The manuscript summarises the methodology behind building computational meshes for spherical shells and provides a useful, novel open-source software to build three commonly used 'hollow sphere' meshes efficiently. Even though there are many nu- merical codes built upon such meshes, I am not aware of any comparable open-source tool to built these meshes from scratch. As such it will proof useful across many numer- ical modelling disciplines in Earth Sciences and for students and advanced researchers alike. The manuscript is concise, well structured and clearly presented. I only have a few suggestions that might make the current manuscript even more useful to interested readers, and a couple of minor specific comments.*

*The manuscript explains the theory behind the code GHOST, but does not explain how to use it in practise. I think it would be useful to most interested readers if the author would consider adding a short section about the actual use of the code. Maybe something similar to section 3 of the manual that is provided with the code itself (which should also be referred to in the manuscript).*

**Section 3 of the manual explains how to run the code in a terminal. GHOST is simply put a mesher. As mentioned in the conclusion of the article, it could be the basis of a FEM or FVM code. The article also showcases an example of how gravity calculations can be carried out on the mesh and how the mesh can be 'filled' with a tomography model, two relevant geophysical applications. Finally the conclusion also mentions additional potential uses.**

*The manuscript provides a nice comparison between three different spherical shell grids, but a conclusive discussion about which grid performs best (which might depend on specific circumstances) is not provided (I guess a grid spacing that is as equal as possible is one criterium). Interested readers might find it useful to read about the author's conclusion on that maybe in the discussion section.*

**I have added a small paragraph on this topic in the conclusion.**

*The author could also consider ensuring the long-term availability of the code by pro- viding a DOI to the code itself (or to a specific version). Zenodo (https://zenodo.org) has, for example, an option to easily link an existing GitHub account and provide a DOI. Also, citing a number (in the title or text) for the specific code version discussed in the manuscript might be helpful.*

**I have generated a v1.0 release and I have linked my github account to Zenodo. This version has been attributed a DOI and it is now explicitely mentioned in the manual and the revised article. https://doi.org/10.5281/zenodo.1245533**

**2  Reviewer2**

*The paper presents open-source software that generates radially projected spherical shell grids, used typically in geophysical applications. Although no novel scientific method is presented, the code will serve well as an exercise for everyone starting with numerical simulations in this field. Unfortunately the source code is only in Fortran, many other modern languages would benefit from such a library as well.*

**This is of course a valid remark but had I written it in Python or C++ such a remark would remain since every modern language has a different way of dealing with objects and would require a different process to interface GHOST to whatever application is being written by the user. Furthermore my PhD student is working on an open source C++ geodynamic code (paper to be submitted in the Fall) which will include a reinvented C++ version of the CITCOM mesh generator.**

*As with every algorithm, complexity is key. From your timing plot in Fig 4 it looks like linear complexity O(n), which would be ideal. If so please make a statement.*

**This is a good remark indeed and I have added a statement in the text in section 2.1.**

*In your introduction you highlight the difficulty to derive the connectivity between cells. I agree and would like to see a chapter on how you tackle this. Another quite similar topic is avoiding duplicate vertices in recursive building algorithms.*

**I have added a few lines of comments to the merge_blocks subroutine entry in the manual. The code is very readable and rather self-explanatory.**

*For numerical simulations it is often important that cells stay as geometrically constant as possible to avoid introducing errors based on mesh irregularities. As you have figured out, a good measure for this error is the area variance of the cells within a shell. Unfortunately you plotted this error in a way that makes selecting the "best" method concerning this error impossible. Also, why the volumetric relative error is of importance in a projected scenario (Fig.7) puzzles me.*

**I have added a second plot to figure 7 which shows the absolute error for all three mesh types.**

*I do not understand what you do in chapter 3. Comparing analytical solutions to numer- ical results is always a good idea. But how do you solve U? What method, order, . . . The nature of your problem suggests a spectral method as others would struggle with the asymptotic boundary condition, please elaborate. Also, as the absolute error within a single shell (lateral only, for i.e. the middle shell) for each cell is interesting in this scenario as it also reflects the sensitivity of the numerical method to mesh irregularities.*

**I have added Eqs 4 and 5 which explicitly explain that gravity vector and potential are computed by means of volume integrals themselves computed using a Gaussian quadrature.**

*In chapter 4 the application is quite useful, but you seem to describe an interpolation method with your statement "I have written a simple interface . . ." . Please describe in detail how you interpolate the data onto the mesh (generic algorithm based on connec- tivity or specific algorithms for each grid type? order of interpolation?). Do not use the word simple.*

**I have reworded these statements and explain how this is done in the second paragraph of section 4.**